# Effect of Low-Immunogenic Yogurt Drinks and Probiotic Bacteria on Immunoreactivity of Cow’s Milk Proteins and Tolerance Induction—In Vitro and In Vivo Studies

**DOI:** 10.3390/nu12113390

**Published:** 2020-11-04

**Authors:** Barbara Wróblewska, Anna Kaliszewska-Suchodoła, Ewa Fuc, Lidia Hanna Markiewicz, Anna Maria Ogrodowczyk, Dagmara Złotkowska, Ewa Wasilewska

**Affiliations:** 1Department of Immunology and Food Microbiology, Institute of Animal Reproduction and Food Research of the Polish Academy of Sciences, Tuwima 10 Str., 10-748 Olsztyn, Poland; b.wroblewska@pan.olsztyn.pl (B.W.); e.fuc@pan.olsztyn.pl (E.F.); l.markiewicz@pan.olsztyn.pl (L.H.M.); a.ogrodowczyk@pan.olsztyn.pl (A.M.O.); d.zlotkowska@pan.olsztyn.pl (D.Z.); 2Institute of Innovation in the Dairy Industry, Kormoranów 1 Str., 11-700 Mrągowo, Poland; anna.suchodola@iipm.pl

**Keywords:** yogurt cultures, *Lactobacillus plantarum*, *Bifidobacterium lactis*, cow’s milk allergy mouse model, desensitization

## Abstract

There is no effective therapy for milk allergy. The role of lactic acid bacteria (LAB) and probiotics in protection against allergy-related outcomes is still under investigation. The aim of the study was to evaluate the immunomodulative and therapeutic potential of yogurt drinks in cow’s milk allergy (CMA) management. We compared immunoreactivity of α-casein (α-CN), β-casein (β-CN), κ-casein (κ-CN), α-lactalbumin (α-LA), and β-lactoglobulin (β-LG) in 27 yogurt drinks fermented with different basic yogurt cultures, or yogurt cultures enriched with *Lactobacillus plantarum* and/or *Bifidobacterium lactis* strains, by competitive ELISA assay. Drinks with the lowest antigenic potential were used as allergoids for CMA therapy. BALB/c mice were sensitized via intraperitoneal injection of α-CN + β-LG mixture with aluminum adjuvant, and gavaged with increasing doses of selected low-immunogenic drinks (YM—basic, or YM-LB—enriched with *L. plantarum* and *B. lactis*) to induce tolerance. Milk- or phosphate-buffered saline (PBS)-dosed mice served as controls. Compared to milk, the immunoreactivity of proteins in drinks increased or decreased, depending on the bacterial sets applied for fermentation. Only a few sets acted synergistically in reducing immunoreactivity. The selected low-immunogenic drinks stimulated allergic mice for profiling Th2 to Th1 response and acquire tolerance, and the effect was greater with YM-LB drink, which during long-lasting interventional feeding strongly increased the secretion of regulatory cytokines, i.e., IL-10 and TGF-β, and IgA and decreased IL-4, IgE, and anti-(α-CN + β-LG) IgG_1_. The studies revealed variations in the potency of yogurt bacteria to change allergenicity of milk proteins and the need for their strict selection to obtain a safe product for allergy sufferers. The YM-LB drink with reduced antigenic potential may be a source of allergoids used in the immunotherapy of IgE mediated CMA, but further clinical or volunteer studies are required.

## 1. Introduction

Cow’s milk allergy (CMA) is common all over the world; however, the development of the industry in the production of novel food focused on people sensitive to milk proteins is limited. Avoiding milk products and all foodstuffs containing milk derivatives that may trigger an allergic reaction is still the only solution for allergists suffering from CMA. In practice, however, this is difficult given the share and role of these proteins in the food industry and the human diet. In addition to standard dairy products, milk proteins are present in a great number of processed foods, such as e.g., meat products, fish products, desserts, or bakery products. So, the exclusion of milk proteins from the diet involves its wide limitation, which leads to negative nutritional, but also social, psychological, and economic effects. The market lacks low-allergenic protein products for allergy sufferers. There is also no treatment for CMA patients. Attempts of oral immunotherapy with milk (OIT) were described, however, most frequently, despite some ad hoc positive results, the meta-analysis of data showed no significant differences between OIT-treated and non-OIT-treated subjects, and that the benefits obtained were counterbalanced by frequent and sometimes serious adverse effects [1,2,3,4]. There is still limited evidence that OIT can induce tolerance or unresponsiveness in CMA sufferers.

Fermentation with lactic acid bacteria (LAB) appeared to be very promising in lowering allergenicity of food compounds and may be a way for therapy of CMA [5,6]. Poza-Guedes et al. [7] demonstrated that daily yogurt intake for six months by children with CMA reduced the disease symptoms and specific IgE levels, as compared to the children with total restriction of cow’s milk-based product. It was also reported that yogurt intake in a total dose of 200 g was tolerated by 64% of the children with systemic cow milk allergy, regardless of the types of adverse effects these children had after consuming cow’s milk [8]. However, many CMA patients do not tolerate commercially available milk-based fermented products, even yogurts [8,9]. According to Food and Agriculture Organization (FAO)/World Health Organization (WHO) standards, yogurt is made of cow’s milk fermented by symbiotic yogurt cultures of *Lactobacillus delbrueckii* subsp. *bulgaricus* and *Streptococcus thermophilus* to the final product containing no less than 10^7^ cfu per gram [10]. There are also bio-yogurts on the market that apart basic yogurt cultures contain other lactobacilli and bifidobacteria with claimed beneficial effect on the human body. Such microbes and their metabolites are able to modulate the intestinal barrier reaction, absorption of nutrients, cytotoxicity, and adverse side effects of digestion in the gut, but their activity depends on the species and very often on the strain, and of course on the health of the body [11,12,13]. Some studies reported a reduction in symptom severity in CMA patients by selected probiotic strains, but they were often criticized for their favored design towards desired outcome and interpretation of the data [14,15]. Others showed that probiotic supplementation increased the risk of allergen sensitization in children with a high-risk of atopic diseases [16,17]. There are also studies showing the allergy-protective effects of sensitive to heating bioactive compounds of milk such as: IgG, lactoferrin, TGF-β, IL-10, alkaline phosphatase (ALP), and osteopontin (OPN), which help to create tolerogenic environment and unresponsiveness upon allergen exposure [18]. They promote regulatory T cells (Tregs) development, induce IgA production, modulate the gut microbiome, and enhance the epithelial barrier. Thus, low-immunogenic milk-based products containing beneficial microbes can be a good solution for people suffering from CMA, however, for allergy patients, bacterial strains as well as whole products should be carefully assessed for their immunoreactivity. Despite possible side effects, rich in calcium and nutrients fermented milk products are still perceived as acceptable dietary supplements for at least some CMA sufferers.

Milk fractionation gives two fractions, caseins and whey. Allergic reactions in children are most often caused by whey proteins, α-lactalbumin (α-LA; Bos d 4) and β-lactoglobulin (β-LG; Bos d 5), and the casein fraction (Bos d 8), whereas in adults, the predominant allergen is casein, and sensitization to whey proteins is rare [19]. Caseins, accounting for 80% of total milk proteins, consists of four fractions: α_s1_-, α_s2_-, β-, and κ-casein (α_s1_-, α_s2_-, β-, and κ-CN), in approximate proportions of 40%, 10%, 40%, and 10%, respectively, of which α_s1_-CN seems to be the most allergenic as it has the highest number of T cell- or IgE-binding epitopes, including some recognized as immunodominant [20,21,22]. Our previous research demonstrated the beneficial effect of bacterial strains from the own collection of the Institute of Animal Reproduction and Food Research of the Polish Academy of Sciences (IAR&FR PAS, Olsztyn, Poland) on whey protein immunoreactivity. The immunoreactive quality of cow’s whey-based drinks was determined in a mouse model [6]. However, the immune recognition of milk allergens depends on the entire food matrix, technological processing, or the presence of accompanying compounds, e.g., carbohydrates. That is why we have now evaluated and compared the suitability of our strains for the preparation of low-immunogenic yogurt drinks from whole milk containing all milk proteins, including highly allergenic and heat-stable caseins. Casein fractions contain different epitopes, which are generally not accessible to antibodies, unless casein decomposes during hydrolysis and digestion [23]. Thus, the selection of bacteria to produce yogurt drinks should be made with the awareness of the possibility of obtaining a product with increased immunogenicity. LAB differ in the release from milk of bioactive peptides with antimicrobial, antioxidant, immunomodulating, opioid, antihypertensive, or cholesterol-lowering properties, and only when properly selected and applied, they improve the quality of pro-health product designated to the appropriate consumer group [5]. A low-immunogenic milk-based product could be used to treat CMA or as a dietary supplement, as a source of protein or peptides. It should be stress that there are no clinical standards for the treatment of food allergy with low-allergenic products, which can act as immunotherapeutic agents. Only some animal models allow observing immunological reactions occurring in the body because of food therapy. The tolerance induction takes place in the gut involving mesenteric lymph nodes (MLNs) and Peyer’s patches (PPs) cells, and net of interleukins which are the components of gut-associated lymphoid tissue (GALT) and splenocytes [24]. We have assumed that yogurt starter cultures differ in their ability to hydrolyse allergenic milk proteins, just like other LAB or bifidobacteria often used for yogurt manufacturing, hence the final effect of such bacteria on the allergenicity of the product obtained also differ. For this purpose, we compared immunoreactivity of α-casein (α-CN), β-CN, κ-CN, α-LA, and β-LG in 27 yogurt drinks fermented with different *L. bulgaricus* and *S. thermophilus* sets, or the same cultures enriched with *Lactobacillus plantarum* and/or *Bifidobacterium lactis* strains. Next, the therapeutic effect of the selected drinks with lowest antigenic potential was determined, in the mouse model with induced allergy to cow’s milk proteins (CMPs). The humoral and cellular response of the mice was monitored over the study.

## 2. Materials and Methods

### 2.1. Bacterial Strains and Growth Conditions

The following bacterial strains were used in the study: *Streptococcus salivarius* subsp. *thermophilus* TKM3, *S*. *thermophilus* MK-10, *S*. *thermophilus* 2K, *Lactobacillus delbrueckii* subsp. *bulgaricus* DB3, *L. bulgaricus* 151, *L. bulgaricus* BK, *Bifidobacterium animalis* subsp. *lactis* Bi30, *B. lactis* J38, *Lactobacillus plantarum* W42, and *L. plantarum* IB. The strains originated from the collection of the IAR&FR PAS (Olsztyn, Poland) and were characterized previously [13,25,26,27]. *S. thermophilus* and *L. bulgaricus* strains were matched in pairs to form starter cultures displaying synergistic effects during the manufacture of yogurts, as follows: *S. thermophilus* TKM3 and *L. bulgaricus* DB3 (set TKM3 + DB3), *S. thermophilus* MK10 and *L. bulgaricus* 151 (MK10 + 151), and *S. thermophilus* 2K and *L. bulgaricus* BK (2K + BK) [28,29,30]. The strains were maintained frozen at −80 °C. For the experiments, strains from frozen stocks were revived and subcultured twice in appropriate nutrient broth using 2% inoculum (*vol/vol*). *S. thermophilus* strains were cultured in M17 broth (Merck KGaA, Darmstadt, Germeny), *L. bulgaricus* and *L. plantarum* strains were cultured in De Man Rogosa Sharpe (MRS) broth (Merck KGaA), and *Bifidobacterium* strains were cultured in Garche’s broth [31]. Incubation was carried out under aerobic or anaerobic conditions (bifidobacteria; 90% N_2_, 5% CO_2_, and 5% H_2_, Bactron600 Anaerobic Chamber, Sheldon Manufacturing Inc., Cornelius, OR, USA) at 37 °C until the stationary phase was achieved (pH~4.5; the growth to about 5–10 × 10^8^ cfu/mL, depending on strain). Finally, for milk inoculation, the active strains were multiplied in skim milk enriched with yeast extract (2 g/L). Yogurt cultures were multiplied together, to pH 4.4–4.5 and the proportion of rods to cocci 1:1–1:2, whereas *L. plantarum* and *B. lactis* strains were multiplied separately, to pH 4.5–4.6. Such activated cultures were directly used for milk fermentation.

### 2.2. Yogurt Drinks

Yogurt drinks were produced using pasteurized skim milk, which contained 3.5% of protein, as determined with the modified Lowry method (Total Protein Kit TP0200; Sigma-Aldrich, St. Louis, MO, USA). Standard yogurt drinks were fermented with the basic yogurt cultures, i.e., *L. bulgaricus* and *S. thermophilus* (three sets described at point 2.1). The remaining drinks were fermented with *L. bulgaricus* and *S. thermophilus* combined with *L. plantarum* and/or *B. lactis* strains. A pasteurized skim milk was used as a control. The exact composition of the bacterial sets used for fermentation is shown in Table 1.

For fermentation, pasteurized milk was prewarmed for 30 min at 37 °C and inoculated with suitable active bacterial strains (2% each, except for 5% for bifidobacteria, *vol*/*vol*). Incubation was carried out aerobically at 37 °C until to final pH 4.6–4.8, i.e., approximately 5–5.5 h. Then, the products were left at room temperature (RT) for 30 min, cooled at 4 °C for 24 h, and used for current experiments. After cooling, the growth of streptococci, lactobacilli, and bifidobacteria was checked as described previously [13,30]. All drinks contained ~10^9^ cfu/g of streptococci, ~10^9^ cfu/g of lactobacilli, and ~1 × 10^8^ cfu/g of bifidobacteria.

### 2.3. Immunoreactivity of the Tested Products

A competitive ELISA was used for assessment of the immunoreactivity of milk and the obtained yogurt drinks with α-CN-, β-CN-, κ-CN-, α-LA-, or β-LG-specific antibodies, obtained from white rabbits as was previously described [32,33]. Lyophilized, antibodies were stored at −20 °C. Before use, the working dilutions of antibodies were titered with indirect ELISA assay.

For immunoreactivity assessment, microplates were coated with 0.5 μg (in a 0.2 M carbonate buffer solution, pH 9.6) per well of α-CN, β-CN, κ-CN, α-LA, or β-LG (Sigma-Aldrich) and left for 1 h at 37 °C. Then, the plates were rinsed three times with PBST buffer (10 mM PBS with 0.5% Tween 20, pH 7.4) and residual free binding sites were blocked with gelatine solution (1.5%) for 30 min at 37 °C. After the washing step with PBST, the wells were filled with 50 μL of sample or standard antigen serial dilution and 50 μL of specific rabbit IgG solution and incubated for 1 h at 37 °C and rinsed with PBST. Next, the wells were filled with goat anti-rabbit IgG conjugated with horseradish peroxidase (A-6154, Sigma-Aldrich) followed by final 1 h incubation at 37 °C and washing step. The color reaction was developed by the 3,3′,5,5′-tetramethylbenzidine substrate (TMB, Sigma-Aldrich, cat. no 5525) for 30 min, then the reaction was stopped, and absorbance was read at 450 nm using a Sunrise spectrometer (Tecan, Grödig, Austria). The immunoreactivity of protein was estimated as a 50% inhibition of antigen binding towards a standard protein [6].

### 2.4. Studies in Mice with Induced Allergy to CMPs

Specific pathogen free (SPF) BALB/c mice (female, 8 weeks old) were obtained from Mossakowski Medical Research Centre of the Polish Academy of Sciences in Warsaw (Poland). Mice were kept in individual ventilated cage systems in the animal facility of the IAR&FR PAS (Olsztyn). Water and diet were provided ad libitum. The diet was free of dairy proteins and was composed to satisfy nutritional requirements of mice [6]. Mice were randomly assigned into groups (*n* = 8/group). Four groups of mice were sensitized via intraperitoneal injection of a mixture of α-CN and β-LG (α-CN + β-LG; 200 μg/100 μL) with aluminum adjuvant (1:1 (*vol*/*vol*); Sigma-Aldrich) on day 0, 7, 14, and 21 (Figure 1). Non-sensitized mice that received intraperitoneally saline served as a negative control for monitoring the immunization process. Starting from day 35 till the end of the experiment on day 63 (for four weeks), experimental feeding of the sensitized mice took place. The mice were gavaged intragastrically with 50, 100, 150, and 200 μL of the tested product during 1st, 2nd, 3rd, and 4th week of feeding, respectively (a dose per mouse per day). The groups were given: S-PBS—0.1 M phosphate-buffered saline, pH 7.4 (PBS); S-M—milk; S-YM—yogurt drink fermented with *S. thermophilus* 2K and *L. bulgaricus* BK (drink YM); and S-YM-LB—yogurt drink fermented with *S. thermophilus* 2K, *L. bulgaricus* BK, *L. plantarum* W42, and *B. lactis* Bi30 (drink YM-LB). Fecal and blood samples were collected once a week all over the study, starting on day 14 after immunization. The weight of the mice and food intake was monitored weekly. The animal care and procedures used in the experiment were approved by the Local Ethical Committee in Olsztyn (43/2015).

#### 2.4.1. Serum and Fecal Sample Collection

Blood samples were taken by puncture of the submandibular vein. After coagulation at RT, they were centrifuged at 3000× *g* at 4 °C for 10 min and serum was collected and stored at −20 °C for further analysis.

For IgA assessment, collected fresh fecal samples were weighted, dissolved 1:10 (*wt*/*vol*) in 0.1 M PBS containing 0.1% NaN_3_ (pH 7.4), homogenized at 4 °C for 10 min (Fugamix, ELMI Ltd., Riga, Latvia) and divided on two parts. One part was centrifuged, and supernatant was collected and frozen at −20 °C for analysis. Dry matter (DM) was determined in the second part. IgA concentration in feces was expressed as amount of antibody per mg of DM.

#### 2.4.2. Indirect ELISA Assay of Total and Specific Antibodies

Total IgE in serum was determined by a commercial Mouse IgE ELISA Set (Cat. No. 555248, BD Biosciences Pharmingen, San Diego, CA, USA). Total IgA in serum or fecal supernatants (obtained as described above) was quantified with Bethyl Mouse IgA ELISA Quantitation Set (E90-103; Bethyl Laboratories, Inc., Montgomery, TX, USA). Assays were performed according to the manufacturers’ instructions.

Indirect ELISA was used for anti-(α-CN + β-LG) IgG_1_ antibodies determination. The 96-well microplates were coated with antigen (α-CN + β-LG mixture) and incubated for 1 h at 37 °C. Following the threefold washing step with PBST, the residual free binding sites were blocked with 1.5% gelatine solution (in coating carbonate buffer, pH 9.6) for 30 min at 37 °C. Then, the plates were washed with PBST, filled with serially diluted sera, and incubated for 2 h at 37 °C. After subsequent washing step with PBST, rabbit anti-mouse IgG_1_ antibodies conjugated with horseradish peroxidase (Cat. No. SAB3701171; Sigma-Aldrich) were added, follow by the applied 1 h incubation at 37 °C and final washing step. The color reaction was developed by incubation with TMB substrate (Sigma, Cat. No. 5525) for 30 min. Then, the reaction was stopped, and absorbance was read at 450 nm using a Sunrise spectrometer (Tecan, Grödig, Austria). Endpoint titer (EpT) values were expressed as the reciprocal of the highest sample dilution that gave a reading about 0.1 OD (optical density) higher that the negative control.

#### 2.4.3. Lymphocyte Isolation

Spleen (SPL) and Peyer’s patches (PPs) were isolated from mice. Tissues were dounced in a chilled incomplete medium (IM; RPMI 1640 supplemented with 1 mM sodium pyruvate, 1 mM non-essential amino acids, and 10 U/mL penicillin-streptomycin), filtered through 80-µm mesh nylon filter (Small Parts Inc., Miami, FL, USA) to remove debris and washed in IM medium. To remove enterocytes, splenocytes were additionally lysed with red cell lysing buffer (Sigma-Aldrich), centrifuged, and then washed away. Finally, the washed cell pellets were suspended in 1 mL of CM medium (RPMI 1640 medium supplemented with 1 mM sodium pyruvate, 1mM non-essential amino acids, 10 U/mL penicillin-streptomycin, and 10% of fetal calf serum) and viability of cells was determined by Trypan blue staining.

#### 2.4.4. ELISpot Assay of IgA and IgG Antibody-Forming Cells (AFC) in Splenocytes

Splenic B cells secreting antigen-specific IgA as well as B cells secreting IgA or IgG irrespective of antigen specificity (total IgA or IgG) were measured using Mabtech ELISpot kits (Mabtech AB, Nacka Strand, Sweden). Total and anti-(α-CN + β-LG) or anti-milk IgA antibody-forming cells (AFC) were assessed using ELISpot for Mouse IgA (Cat. No. 3835-2HW-Plus) and total IgG AFC were assessed using Elispot for Mouse IgG (Cat. No. 3825-2HW-Plus). Assays were performed according to the manufacturers’ procedures. Briefly, MultiScreen-IP Filter plates (MSIPN4510, Merck Millipore Ltd., Carrigtwohill, Ireland) were activated with 70% ethanol, washed five times with sterile water, coated with 100 µL of anti-mouse IgA (10 µg/mL), anti-mouse IgG (15 µg/mL), mixture of α-CN + β-LG (50 µg/mL in PBS, pH 7.4) or milk and incubated overnight at 4 °C. Next, the plates were washed with washing buffer (PBS with 0.1% Tween 20, pH 7.4) and residual free binding sites were blocked using CM medium (200 µL/well) for 30 min at RT. After that, freshly isolated lymphocytes from the spleen were added at amount 1 × 10^6^ cells/well for specific and at 1 × 10^5^ cell/well for total AFC assessment and plates were incubated in a 5% CO_2_ atmosphere at 37 °C for 24 h and washed. Plates were then incubated with 100 µL/well of biotin anti-IgA or anti-IgG (1 µL/mL in PBS with 0.5% fetal calf serum—PBS-0.5% FCS) and incubated overnight at 4 °C in a humidity chamber. After following the washing step, plates were incubated at RT for 1 h with 100 µL/well of PBS-0.5% FCS containing streptavidin HRP diluted 1:1000. Finally, plates were washed as previously, and color reaction was developed by the 3-amino-9-ethyl-carbazole (Moss Inc., Pasadena, MD, USA). The plates were incubated at RT for 15–60 min until spots developed, and thereafter washed with tap water. Spots were read the next day using a stereo zoom microscope (SZX9, Olympus, Tokyo, Japan). Results were presented as the number of AFC for 10^6^ cells.

#### 2.4.5. Cytokine Secretion Assay

IFN-γ, TGF-*β*, IL-4, and IL-10 were assessed using commercial mouse cytokine ELISA sets. BD OptEIA^TM^ Mouse ELISA sets (BD Biosciences Pharmingen, San Diego, CA, USA) were used for: IFN-γ (Cat. No. 555138), IL-4 (Cat. No. 555232) and IL-10 (Cat. No. 555252), and eBioscience ELISA set for TGF-β (Cat. No. 88-8350; eBioscience, Inc., San Diego, CA, USA). Freshly isolated lymphocytes from SPL or PPs and suspended in CM medium were added into 48-well cell culture plates at a number of 1 × 10^6^ cells/per well and incubated in a 5% CO_2_ atmosphere at 37 °C for 24 h. Thereafter, for stimulation, 20 μL of milk or the mixture of α-CN + β-LG (1 mg/mL in PBS) was added to each well. Cells cultured without stimulant served as a control. After following 48 h of incubation at 37 °C in a 5% CO_2_ atmosphere, the cells were centrifuged at 400× *g* for 10 min at 4 °C (Eppendorf 5804 R, Hamburg, Germany) and supernatants were collected and stored at −80 °C for cytokines analysis. Procedures for cytokines concentrations assessment were performed according to the manufacturers’ instructions.

#### 2.4.6. Characteristics of Gut Microbiota

Bacterial DNA was isolated from fresh cecal contents (c.a. 0.1 g) using GeneMATRIX Stool DNA Purification Kit (EURX, Gdansk, Poland) and a bead-beating method recommended by the manufacturer. Until analysis, extracted DNA was stored at −20 °C. Polymerase chain reactions were performed as described previously [6] using primers universal for bacteria (968-GC-f: GC clamp-aacgcgaagaacctta, 1401-r, cggtgtgtacaagaccc (where GC clamp was cgcccggggcgcgccggcggcggcccgggggcccggggg) and specific for lactobacilli (Lacsspf-GC: GC clamp-ccaccgttacaccgggaa, Lacsspr: ccattgtggaagattccc). Reaction mixtures (30 µL) contained 3 µL of 10× concentrated reaction buffer, 1 U of *Taq* DNA polymerase (Thermo Fisher Scientific, Waltham, MA, USA), each dNTP at 200 µM, each primer at 250 µM, magnesium chloride at 5 mM, 1 µL of DNA and sterile water. Thermal profile of amplifications included a 5 min denaturation step followed by 35 cycles of denaturation (30 s, 95 °C) annealing (30 s, at the temperature of 57 or 58 °C, respectively) and elongation (30 s, 72 °C), and the reaction was finalized by the final elongation at 72 °C for 20 min.

PCR products were separated in polyacrylamide gels (8%, acrylamide: bisacrylamide 37.5:1) with a gradient of denaturants (7 M urea and 40% *v*/*v* formamide). The denaturing gradient gel electrophoresis (DGGE) technique was applied with denaturing gradients ranging from 30% to 75% for eubacteria and 40–55% for lactobacilli. Electrophoresis was conducted in the DCode Universal Mutation Detection System (Bio-Rad Laboratories, Inc., Hercules, CA, USA) in 0.5× TAE buffer at 200 V for 10 min and 85 V for 20 h. After electrophoresis, gels were stained with 10,000× diluted SYBR Green I (Sigma-Aldrich) for 10 min and then photographed under UV light (Image XR system, Bio-Rad). Selected bands were cut off from the gel and re-amplified with the same primers, purified with the PCR/DNA Clean-Up Purification Kit (EURX, Gdansk, Poland) and commercially sequenced at Genomed (Warsaw, Poland). Identification of bands was based on sequence similarity to the sequences deposited in GenBank and was performed using the BLASTn algorithm.

To assess the changes in microbiota structure, the similarity of DGGE profiles was calculated using the Pearson’s product-moment correlation coefficient and presented as a dendrogram constructed using the unweighted pair group method with arithmetic mean (UPGMA) (BioNumerics software, Applied Maths, Sint-Martens-Latem, Belgium).

### 2.5. Statistical Analysis

Results were expressed as means ± SD. The *t* test was used to evaluate differences between two groups, whereas one-way ANOVA followed by post hoc Tukey test were applied for multiple comparisons. The Kruskal–Wallis test was also used—for multiple comparison of nonparametric data. The results were statistically significant if *p <* 0.05.

## 3. Results and Discussion

### 3.1. Immunoreactivity of Yogurt Drinks Fermented with Different Bacterial Sets

LAB and *Bifidobacterium* species and strains differ in their proteolytic activities [34]. In addition to converting lactose to lactic acid and thus acidifying the environment, which spontaneously changes the protein structure, bacterial proteases hydrolyze proteins to peptides and free amino acids, which can have a direct effect on the destruction of epitope structures, mainly conformational but also linear, and thus on the immunoreactivity of proteins. Figure 2 and Figure 3 show the immunoreactivity of whey and casein proteins, present in milk and in the tested yogurt drinks, with anti-α-LA and anti-β-LG antibodies and anti-α-CN, anti-β-CN, and anti-κ-CN antibodies, respectively.

Compared to milk, all basic yogurt cultures (TKM3 + DB3, MK10 + 151, and 2K + BK) lowered α-LA and β-LG immunoreactivity (up to 4.4-times and 10-times for α-LA and β-LG, respectively; *p* < 0.05 in comparison to milk; Figure 2), and the reducing effect, for both α-LA and β-LG, was greater for MK10 + 151 set and 2K + BK set in comparison to TKM3 + DB3 set (*p* < 0.05, the differences marked with letters, and the compared values are shown in Figure 2A–C and Figure 2D–F for α-LA and β-LG, respectively). The enrichment of basic yogurt sets with *L. plantarum* strain (IB or W42) or/and *B. lactis* strain (Bi30 or J38) changed fermentation terms, and consequently, the allergenic potential of the obtained yogurt drinks. The α-LA immunoreactivity was generally reduced by adding *L. plantarum* (up to 2.5-times), whereas, enhanced by *B. lactis* strains (up to 1.8-times), and the final effect depended on the applied basic set (*p* < 0.05 in comparison to basic set, column “alone”; Figure 2A–C). As regards to β-LG, *L. plantarum* W42 in combination with TKM3 + DB3 or MK10 + 151 decreased reactivity of this protein, as did the Bi30 strain in combination with TKM3 + DB3 (*p* < 0.05 vs. alone starter set). Despite this, when added to MK10 + 151 or 2K + BK set, the *B. lactis* Bi30 strain increased β-LG immunoreactivity, similarly as did the strain *B. lactis* J38. However, the final effect was still 3.6-5-times reduced compared to milk (*p* < 0.05). Opposite to the *L. plantarum* W42, we did not observe any effect of the *L. plantarum* IB on β-LG immunoreactivity. Kleber et al. [35] described the strain-specific ability of LAB to weaken β-LG antigenicity and the synergism in antigens reduction by *S. thermophilus* in combination with some lactic acid bacteria. Moreover, some LAB strains were more effective in milk, and other in sweet whey. Our results indicate that apart from bacteria, the chemical composition and protein matrix of the raw material may affect the final immunoreactivity of the fermented product. The immunoreactivity of whey proteins fermented with 2K + BK or 2K + BK + Bi30 + W42 set was lower in whey drinks, as compared to yogurt drinks [6]. In milk, the combined effect on the α-LA and β-LG immunoreactivity of the four bacteria used in fermentation (*S. thermophilus*, *L. bulgaricus*, *L. plantarum*, and *B. lactis*) was higher, lower than or equal to that observed for single strains of *L. plantarum* or *B. lactis* combined with primary yogurt cultures. In some cases, the previously observed increase in immunoreactivity caused by one of the strains was fully or partially abolished by the strain that showed a reducing effect. Concerning α-LA, the Bi30 + W42 and J38 + W42 combinations were generally effective with all basic starter sets, but mainly due to the combined activity of the W42 strain and yogurt cultures (*p* < 0.05 vs. both standard drinks and milk; Figure 2A–C). Similarly, β-LG immunoreactivity was effectively reduced by the Bi30 + W42 pair, but only in the combination with TKM3 + DB3 set. We observed increased β-LG immunoreactivity in the yogurt drinks fermented with Bi30 or J38 combined with *L. plantarum* and MK10 + 151 or 2K + BK starters, caused mainly by the *Bifidobacterium* strains, as no additional effect was observed for W42 or IB added to these sets (*p* < 0.05; Figure 2F). Even so, the immunoreactivity of this protein remained still effectively reduced compared to milk (3.3-times). A much more enhancing effect was observed for α-LA and the combination of the J38 strain with the *L. plantarum* IB and 2K + BK yogurt set.

The composition of bacterial sets used for milk fermentation differently impacted the immunoreactivity of casein proteins (Figure 3). As to regards basic sets, α-CN immunoreactivity increased in the TKM3 + DB3 fermented drink but decreased in milk fermented with MK10 + 151 and 2K + BK (*p* < 0.05 vs. milk; Figure 3A–C). Alike, the binding capacity of β-CN-specific antibodies increased in the TKM3 + DB3 drink and decreased as a result of activity of MK10 + 151 (*p* < 0.05 vs. milk), but it was non-affected by the 2K + BK set (Figure 3D–F). In turn, κ-CN epitopes did not change after fermentation with the TKM3 + DB3 set, but their affinity to specific antibodies increased due to application in the process of the MK10 + 151 set and decreased due to the 2K + BK set (*p* < 0.05; Figure 3G–I). The addition of *L. plantarum* or *B. lactis* strains had no influence or increased immunoreactivity of α-CN in drinks containing TKM3 + DB3 strains and decreased in drinks containing MK10 + 151 or 2K + BK strains (Figure 3A–C). Except for the IB + MK10 + 151 set, the tested *L. plantarum* or *B. lactis* strains combined with MK10 + 151 or 2K + BK starters effectively reduced α-CN immunoreactivity (*p* < 0.05 in comparison to standard drinks and milk; Figure 3B,C). Though, the α-CN immunoreactivity was the most effectively reduced by the Bi30 + W42 pair in a set with MK10 + 151 or 2K + BK (*p* < 0.05 compared to basic sets and milk; Figure 3B,C). Regardless, the *L. plantarum* W42 added to the TKM3 + DB3 increased affinity of the tested proteins to α-CN- and β-CN-specific antibodies (up to 2.8-times and 4.4-times, respectively; *p* < 0.05 vs. alone basic set and milk; Figure 3A,D). However, we observed synergistic action of this strain in combination with bifidobacteria and some yogurt cultures. Despite that the W42 strain increased immunoreactivity of α-CN and β-CN in the milk fermented with TKM3 + DB3, and the Bi30 had no influence, the combination of these strains significantly decreased immunoreactivity of the proteins (*p* < 0.05 compared to alone TKM3 + DB3 set and milk; Figure 3A,D). The same was true for the J38 strain and α-CN, probably due to synergistic activity of all or some of these bacteria. On the contrary, compared to the baselines, β-CN immunoreactivity was increased by both *B. lactis* strains added to MK10 + 151 (*p* < 0.05), although ultimately the effect was still the same as in milk (Figure 3E). The Bi30 strain was the only one to reduce β-CN immunoreactivity in combination with the 2K + BK set (*p* < 0.05 vs. milk and standard 2K + BK drink; Figure 3F). The influence of the tested bacterial sets on κ-CN was much more varied. Through synergistic action, the Bi30 strain when combined with W42 and TKM3 + DB3 reduced κ-CN immunoreactivity, which was not the case with single strains added to these yogurt cultures (*p* < 0.05 vs. basic set and milk). Similar immunomodulatory effect exerted J38 + W42 and J38 + IB pairs of strains when added to TKM3 + DB3 and MK10 + 151 sets, however, mostly due to the activity of the J38 strain (*p* < 0.05 vs. standard drinks and milk; Figure 3A,B). Compared to basic sets, *L. plantarum* and *B. lactis* strains increased κ-CN immunoreactivity when combined with 2K + BK, whereas the Bi30 strain when combined with MK10 + 151 (*p* < 0.05; Figure 3I). Promisingly, *L. plantarum* W42 eliminated the booster effect caused by the Bi30 strain in combination with MK10 + 151 and 2K + BK sets (*p* < 0.05 vs. standard yogurt drinks and milk; Figure 3B). However, no synergistic effect was identified for *B. lactis* J38 combined with *L. plantarum* IB and 2K + BK yogurt cultures. Rui et al. [36] described eight *L. plantarum* strains showing potency in reducing IgE reactivity with soy proteins isolates by 83.8–94.8%, as determined by ELISA assay. In turn, Snel at al. [37] described strain-specific immunomodulatory effects of *L. plantarum* strains on birch-pollen-allergic subjects. In our study, the IB strain was generally ineffective against casein proteins, but when used with the 2K + BK set, it decreased α-CN immunoreactivity but increased κ-CN immunoreactivity (*p* < 0.05 vs. both alone 2K + BK set and milk; Figure 3C,I). The κ-CN appears to be the protein whose immunoreactivity is most difficult to reduce during the fermentation process. Our recent studies showed that κ-CN has a similar immunogenic potential to α-CN [38]. Moreover, its immunoreactivity may further increase during intestinal digestion [39]. Casein is rapidly and extensively degraded by proteolytic enzymes as it has an open and flexible structure, with peptide bonds easily exposed to enzymatic digestion. It is assumed that caseins are responsible for the strongest systemic allergic reaction as they have the immunodominant epitopes, most frequently recognized by the IgE of CMA patients [19,22]. The development of a set of strains that effectively reduce the immunoreactivity of these proteins can be a distinctive feature of safe food for allergy sufferers.

The experiment showed variations in the potency of yogurt starter cultures to change the allergenicity of milk proteins, as well as the need of for strict selection of bacteria to obtain safe yogurt drinks for allergy sufferers. This may be causally related to the different yogurt tolerance by CMA sufferers but will require further clinical studies. Not all enriched bacterial sets exert synergistic effect in lowering milk protein immunoreactivity. Although a significant decrease in immunoreactivity caused by some yogurt cultures was observed, after enrichment with other LAB or *Bifidobacterium* strains, immunoreactivity often increased. Besides, the effect was varied for different milk proteins.

Since the Bi30 + W42 + 2K + BK set was effective in reducing the immunoreactivity of most milk proteins, we used it to make the yogurt drink for further in vivo studies in a mouse model of CMA. Previously, we used this set to study the immunoreactivity of fermented whey and the results obtained in the CMA mice were very promising [6]. However, due to meaningful residual immunoreactivity of casein proteins present in the yogurt drink obtained with Bi30 + W42 + 2K + BK set, there was a need for further research on its activity in the whole milk matrix, in the presence of casein proteins. Besides, it was the only set that effectively reduced β-CN immunoreactivity (Figure 3D–F).

### 3.2. Animal Studies

In allergy, a shift of Th1/Th2 balance towards the pre-allergic Th2 response takes place. We applied an experimental feeding model—the administration of increasing doses of allergens, as a kind of allergen immunotherapy, in CMA mice, using the selected yogurt drinks as allergoids, whose aggressiveness was reduced by fermentation, so that it could change the Th2- towards Th1-mediated immune reaction, and induce tolerance. This model was successfully used in our previous study [6]. The BALB/c mice when stimulated with antigens give Th2 type response with IgE secretion, which is crucial for the pathomechanism of immediate allergic response [6,40]. The high titer of IgE and IgG_1_ specific antibodies and the elevated IL-4 level are recognized mediators of inflammation during IgE-dependent allergic reaction in both humans and experimental animals [41,42]. On the 35th day of the experiment (i.e., before starting the experimental feeding), the mouse humoral responses displayed that the applied administration regimen and α-CN + β-LG doses used for mice immunization increased serum IgE level (about 8-fold, from 72 up to 600 ng/mL) and anti-(α-CN + β-LG) IgG_1_ titer (the EpT above 2^10^), thus demonstrating an immune response adequate to the major milk allergens due to IgE-mediated hypersensitivity to CMPs [6,43]. Besides, the α-CN + β-LG sensitized mice showed increased secretion of IL-4 by PPs and SPL, and lowered IL-10 and TGF-β secretion in PPs but increased in SPL [6]. Two weeks after the last antigen challenge, when allergy response fully developed and stabilized, we started experimental feeding with YM (fermented with 2K + BK set) and YM-LB (fermented with Bi30 + W42 + 2K + BK set) yogurt drinks in order to induce suppression of a cellular and humoral immune response to milk antigens. Sensitized mice fed with milk (S-M group) or PBS (S-PBS group) served as controls. The body weight of the mice and food intake was monitored over the study. No significant differences were observed between the groups in the body weights or food intakes when measured at the same time points (Figure 4). In addition, total food intake during the study was not different for any of the tested groups. However, there was an increase in delta body weight over the entire study period in S-YM and S-YM-LB groups compared to S-M and S-PBS groups (*p* < 0.05; Figure 4A).

#### 3.2.1. Humoral Response of Sensitized Mice to Yogurt Drinks

The administration of increasing doses of the tested yogurt drinks to the sensitized mice caused a calming down of the allergic reaction. The allergic mice on a milk-free diet (S-PBS group) could not reduce the amount of circulating antibodies by themselves (Figure 5). In turn, after a month of treatment, the serum IgE increased in the S-M group administered milk (*p* < 0.05 in comparison to the sensitized but PBS-receiving mice; Figure 5A). In the S-M group, milk proteins were still recognizable by the body as strong allergens and stimulated the immune system to continuous production of IgE. Conversely, the IgE level decreased nearly twice in the serum of YM or YM-LB fed mice, as compared to the control mice receiving PBS and three times in comparison to the S-M group receiving milk (*p* < 0.05; Figure 5A). Similarly, anti-(α-CN + β-LG) IgG_1_ decreased in the two groups of mice receiving yogurt drinks (*p* < 0.05 vs. S-PBS and S-M groups; Figure 5B), which shows an immunotherapeutic effect of YM and YM-LB yogurt drinks. IgG_1_ is the first class of G immunoglobulin to be formed because of prolonged exposure to the antigen. It is even presumed that IgG_1_, like IgE, may induce systemic anaphylaxis [44]. Hence, the obtained reduction of IgE and IgG_1_ antibodies may predict further, beneficial changes in the animal’s immune system and consequently induce tolerance to allergens. Shek et al. [45] showed that lowering the specific IgE level during the first years of illness may result in a higher probability of achieving a natural state of tolerance in humans. Thus, considering the humoral response of the tested mice (decreased IgE and IgG_1_ levels), the usefulness of both yogurt drinks for oral immunotherapy was demonstrated. Furthermore, the S-YM-LB group showed a significant increase in total IgA in serum (*p* < 0.05 vs. control PBS receiving mice; Figure 5C), and the S-YM group in feces (*p* < 0.05 vs. other groups; Figure 5D). Secretory IgA (sIgA) is one of the principal defense mechanisms of the mucosa-associated lymphoid tissue to specific antigens encountered along all mucosal surfaces. It protects the epithelium, by blocking the receptors prevents bacterial adhesion and absorption of antigens or potentially allergenic molecules. By coating the allergen, sIgA blocks its connection with IgE and, consequently, may contribute to calming down the allergic reaction, mitigating the inflammation and reducing the intestinal hypersensitivity reaction [46]. The evidence exists on the anti-inflammatory role of dimeric IgA (dIgA), due to the intracellular neutralization of bacterial antigens (e.g., LPS) involved in the proinflammatory activation of the intestinal epithelial cells [47]. Frossard et al. [48] compared the IgA antigen-specific cells in PPs of mice with induced β-LG allergy and those in mice with induced food tolerance to the same proteins by means of appropriate dietary protocols. Mice that developed a tolerance state had a higher number of anti-β-LG IgA positive cells in PPs, and a higher IgA titer in the intestinal contents compared to mice with symptoms of β-LG anaphylaxis. The Elispot assay revealed that both YM and YM-LB drinks stimulated IgA production in splenocytes (Figure 5E). However, the number of IgA AFC in splenocytes of S-YM-LB mice differed from both S-PBS and S-M groups (*p* < 0.05), whereas the number of IgA AFC in splenocytes of S-YM mice differed only from S-PBS group (*p* < 0.05). There were no differences in anti-milk and anti-(α-CN + β-LG) IgA AFC between the mice groups (Figure 5F,G). Mice that received the YM-LB drink were characterized by a significantly elevated level of total IgG AFC (*p* < 0.05 vs. both control groups; Figure 5H). So, the YM-LB drink was a better stimulator for both IgA and IgG production. Increased total IgG response in mice from groups receiving fermented drinks may indicate a natural response of the body to the bacteria applied in the process. In turn, an increased number of cells producing G-class antibodies results in the further activation of the immune system cells, and finally enhances IgA production. Weiberg et al. [49] demonstrated that the spleen plays a major role in the gut immune response, serving as a reservoir of immune cells that migrate to the site of antigen entrance. It appeared that stimulation of recipient mice by orally administered antigens enhanced the migration of the splenic B cells into the gut as well as their switch to IgA+ plasma cells.

#### 3.2.2. Cytokine Release in Splenocytes and Peyer’s Patches Cultures of the Tested Mice

Cytokines are the mediators of balance of Th1 and Th2 immune response, and control IgE synthesis. During the induction of tolerance, the administration of small doses of antigen results in the generation of antigen-specific T-cells which secrete cytokines that regulate the immune response, i.e., TGF-β or IL-10 [50]. IL-10 and TGF-β are anti-inflammatory and regulatory cytokines associated with the phenomenon of tolerance and control of immune processes manifested as inflammation, infection, or allergies [51,52]. Homeostasis disorders resulting from Treg cell deficiency in the intestine are believed to cause an increased Th2 response in allergic diseases. Different types of T regulatory lymphocytes (Tregs), such as Foxp3^+^ Tregs, lymphocytes Tr1 releasing IL-10, and lymphocytes Th3 releasing TGF-β are associated with oral tolerance [53]. Similarly, Breg-mediated suppression is an important means for the maintenance of peripheral tolerance that appears to be directly mediated by the production of IL-10 and/or TGF-β and by the ability of B cells to interact with pathogenic T cells to inhibit harmful immune responses [54]. In the gastrointestinal tract of mice, there is a subpopulation of dendritic cells characterized by the expression of the surface particle CD103, which participate in the formation of food tolerance by strong induction of Foxp3 factor in CD4^+^ lymphocytes in the presence of TGF-β and vitamin A metabolites [55]. It has been shown that an increase in TGF-β secretion in splenocyte cultures may indicate the beginning of the process of excitation of regulatory cells.

Our results showed that the applied therapy, i.e., the administration of increasing doses of low-immunogenic yogurt drink YM or YM-LB to allergic mice increased the secretion of IL-10 and TGF-β by PPs and SPL in response to the in vitro challenge with α-CN + β-LG (*p* < 0.05 in comparison to S-PBS and S-M groups; Figure 6). However, the cytokines secretion was significantly greater in the culture of PPs from YM-LB fed mice (*p* < 0.05 as compared to S-YM group; Figure 6A,D). In addition, the splenocytes of mice receiving the YM-LB drink secreted much more IL-10 in response to stimulation with milk (*p* < 0.05 vs. other groups; Figure 6F). Thus, both yogurt drinks, YM (fermented with *S. thermophilus* 2K and *L. bulgaricus* BK) and YM-LB (fermented with *S. thermophilus* 2K and *L. bulgaricus* BK combined with *L. plantarum* W42 and *B. lactis* Bi30), contributed to the production of regulatory cytokines, however, the secretion of these cytokines was much greater in the case of YM-LB drink, where four strains were involved in the fermentation process. We obtained similar tendency while feeding sensitized BALB/c mice with whey fermented with these bacteria, which confirms their beneficial effect on CMA [6]. The administration to sensitized mice of milk or PBS did not cause meaningful differences in the secretion of IL-10 by PPs and SPL cells, after stimulation in culture with α-CN + β-LG or milk. Furthermore, in the SPL culture from mice receiving milk, the secretion of IL-10 was further decreased after exposure to α-CN + β-LG (*p* < 0.05 vs. other groups; Figure 6E). That regulatory cytokine was effectively induced in the PPs of YM and YM-LB administered mice, in place where the first contact between allergens and the body took place. PPs play a role in the recognition and production of an immune response to antigens. T cells, B cells, and memory cells are stimulated upon encountering an antigen in Peyer’s patches. We observed significant increase of the secretion of immunoregulatory cytokine IL-10 because of administration of the tested yogurt drinks. Especially, the one containing *L. plantarum* and *B. lactis* strains appeared to be effective in modulating the immune response of mice with CMA towards acquiring tolerance. In one of our previous studies, we observed an increased input of CD3^+^CD4^+^CD25^+^ regulatory T cells combined with increased IL-10 secretion in splenic lymphocytes of BALB/c mice with DSS-induced colitis, when treated with *S. thermophilus* MK-10 and *L. bulgaricus* 151 strains (MK10 + 151 set) [13]. Sierra et al. [51] demonstrated the ability of a probiotic strain of *L. salivarius* to activate the immune system by increasing the secretion of IL-10 and TGF-β. In addition, *Bifidobacterium longum* and *L. plantarum* strains increased the secretion of IL-10 in some studies [37,56]. However, it should be stressed that the immunomodulating effect was a specific feature of some bacterial strains. LAB differed in their ability to induce regulatory Foxp3^+^ T cells in vitro, and the strains that induced these cells also caused increased secretion of IL-10 [57].

INF-γ is one of the main cytokine, next to IL-2 and TNF-α, produced by the subclass of Th1 lymphocytes, which may reduce the increased Th2 response in allergic diseases by influencing, among other, the regulation of IL-4 level and the increase of scavenger activity of different cells. IL-4 plays a key role in the pathomechanism of food allergy by activating plasma cells to produce E class antibodies. IFN-γ produced by Th1 cells and IL-4 produced by Th2 counter-regulate each other. In our model, however, no significant changes in IFN-γ secretion in PPs and SPL after sensitization of naïve mice were observed, although a significant increase in IL-4 was noted [6]. However, we have not examined the IFN-γ activity in the lamina propria, where these changes would probably be better noticeable. We found some differences in INF-γ levels in cultures of lymphocytes originating from the tested groups of mice and exposed to allergens in vitro. INF-γ was reduced in the cultures of lymphocytes isolated from PPs of S-YM and S-YM-LB groups, and from SPL of S-M, S-YM, and S-YM-LB groups during stimulation with α-CN + β-LG, as compared to S-PBS group (*p* < 0.05; Figure 6G,H). Of the groups which stayed on milk proteins (S-M, S-YM, S-YM-LB), the cells isolated from PPs of the YM fed mice produced less IFN-γ in response to stimulation with α-CN + β-LG, while the cells isolated from SPL of these mice increased INF-γ secretion as a result of stimulation with whole milk proteins (*p* < 0.05 vs. other groups; Figure 6G,I). Thus, in this case, only the YM drink influenced the IFN-γ production in PPs and SPL. In the mid-20th century, IFN-γ was considered as the main marker of the Th1 response, playing the role of inhibiting the Th2 response and allergy [58]. Today, the role of IFN-γ remains debatable, although it is pointed out that IFN-γ can play a protective role against an allergic reaction [59]. Perrier et al. [60] demonstrated that despite high levels of IFN-γ secreted by cells of spleen and mesenteric lymph nodes after allergen stimulation in cultures, IFN-γ did not inhibit the development of a severe allergic reaction in sensitized animals. Thus, IFN-γ presence cannot be considered a universal protection mechanism. In our study, based on the IFN-γ level, we cannot clearly state that the applied therapy with YM and YM-LB yogurt drinks stimulated the immune system cells to acquire the Th1 response. However, both yogurt drinks decreased IL-4 secretion as compared to the S-PBS group, on a milk-free diet, regardless of origin of lymphocytes and the way of their stimulation (*p* < 0.05; Figure 6J–L). In splenocyte cultures isolated from S-YM and S-YM-LB groups, the concentration of IL-4 due to α-CN + β-LG stimulation was lowest and differed from both S-PBS and S-M groups (*p* < 0.05; Figure 6K). Thus, the tested yogurt drinks mitigated inflammatory reactions caused by α-CN + β-LG allergy in mice. Most studies on the antiallergic activity of LAB indicate that LAB induce Th1 response and simultaneously inhibit Th2 and IgE production [61]. However, Schabussova et al. [62] demonstrated that the administration of *L. paracasei* or *B. longum* strains to poliallergic mice resulted in a generalized immunosuppression of T-cell response, rather than a unidirectional change of the Th2 responses to the Th1 phenotype. Our findings displayed that the tested yogurt drinks, especially YM-LB, activate protective regulatory mechanisms and induce a shift of the allergen-specific Th2 response towards the Th1 phenotype, however, other Th1 mediators than IFN-γ may be engaged in the process. As a result of reduced activity of Th2 cells, the secretion of IL-4 by splenocytes of YM and YM-LB-administered mice decreased, which was accompanied by a decrease of IgE and anti(α-CN + β-LG)G_1_ levels in serum. At the same time, a significant increase in the concentration of regulatory cytokines IL-10 and TGF-β was noted, both in PPs and SPL cultures, similarly as the IgA concentration in serum and feces. Mice receiving yogurt drinks had higher body weights.

#### 3.2.3. Effect of the Tested Yogurt Drinks on Cecal Microbiota

We performed a qualitative and semi-quantitative analysis of eubacteria and *Lactobacillus* populations using the PCR-DGGE method, which allows for characterization of bacterial groups that predominate in complex populations. The analysis of DGGE profiles reflecting total bacteria (eubacteria) did not reveal a clear effect of the tested drinks on the structure of the mouse cecal microbiota, however, some treatment-specific grouping can be observed (Figure 7A). DGGE patterns of microbiota of mice fed yogurt drinks (S-YM and S-YM-LB groups) were grouped into one cluster at minimal similarity of 58% (Figure 7A; cluster I), whereas control groups were separated into cluster with two subclusters comprising of most of mice from S-M and S-PBS groups (Figure 7A; cluster II and III, respectively). Unfortunately, attempts of sequencing and identification of single DGGE bands failed probably due to their proximity and inability to obtain clear reads. A dendrogram obtained for *Lactobacillus* DGGE patterns showed that control mice (S-PBS) were predominantly separated from the milk and yogurt-fed ones, and their overall similarity was over 85% (Figure 7B, cluster I). In addition, with one exception, the DGGE patterns of the *Lactobacillus* population of S-M mice were grouped separately from the patterns of YM- and YM-LB-fed mice (Figure 7B; cluster II and cluster III, respectively), although they all showed high (about 94%) similarity. Thus, the presence of milk and yogurt cultures in the diet of mice slightly impacted the *Lactobacillus* population. The sequencing of selected DGGE bands from *Lactobacillus* DGGE banding patterns revealed the presence of *Lactobacillus reuteri*, *Lactobacillus gasseri*, *Lactobacillus murinus*, and *L. delbrueckii* spp. *bulgaricus* in the gut of the tested mice. The first three species belong to a typical mouse microbiota, while *L. bulgaricus* was detected in an individual from the group that received the YM-LB drink. The performed analysis indicates that the bacteria present in yogurt drinks survived in the gut but did not tend to overgrow the autochthonous microbes. Despite the long-lasting feeding, the intestinal microbiota was a strong competition for the tested bacteria, preventing them from increasing their population in the intestine. Mazmanian et al. [63] revealed that introduction of *Bacteroides fragilis* into the gut of germ-free mice led to a redevelopment of GALT and induction of tolerance if performed in the neonatal period. Our mice were already inhabited by microbiota, so it was difficult to observe meaningful changes. However, evidence exists, that gut microbiota effects the development of Treg cells and tolerance induction [64]. Besides, a positive correlation was found between the numbers of Gram-positive anaerobes (lactobacilli and bifidobacteria) and Gram-negative anaerobes (*Bacteroides* and *Prevotella* species) in maternal stool and IL-10 secretion by cord blood mononuclear cells [65].

## 4. Summary

The composition of bacterial sets applied in the production of yogurt, both standard and bio-yogurt enriched with other LAB or bifidobacteria, can exert a profound effect on milk protein immunoreactivity, especially caseins, and should therefore be controlled in yogurts intended for CMA patients. Compared to milk, both an increased and decreased protein immunoreactivity in the yogurt drinks tested was observed, depending on the applied bacterial set. Only some of the basic yogurt cultures enriched with lactobacilli and/or bifidobacteria acted synergistically and further reduced immunoreactivity of milk proteins compared to standard sets. Stimulation of the body allergic to cow’s milk proteins with yogurt drink YM or YM-LB with allergoid characteristics led to the profiling of Th2 to Th1 response, indicating stimulation of tolerance process. We noted a significant enhance of secretion of IL-10 and TGF-β regulatory cytokines by PPs and SPL of YM- and YM-LB-administered mice. Simultaneously, as a result of decreased activity of Th2 cells, the secretion of IL-4 by splenocytes decreased, as did IgE and anti-(α-CN + β-LG) IgG1 antibodies in serum. In turn, there was an increase in total IgA secretion in serum and feces, and total IgA and IgG AFC in splenocytes. Such a mechanism accompanies the immune response restoring the balance from Th2 into Th1. So, the analyzed markers, taken together, provided a compendium of results that the yogurt drink YM-LB (fermented with *S. thermophilus* 2K and *L. bulgaricus* BK combined with *L. plantarum* W42 and *B. lactis* Bi30 strains), as showing a low antigenic potential, may be a source of allergoids used in the immunotherapy of IgE mediated cow’s milk allergy. However, this should be confirmed in further clinical studies or involving human volunteers. Future studies should also be aimed at analyzing the presence of bioactive peptides supporting the immune system in particularly vulnerable groups of people.

## Figures and Tables

**Figure 1 nutrients-12-03390-f001:**
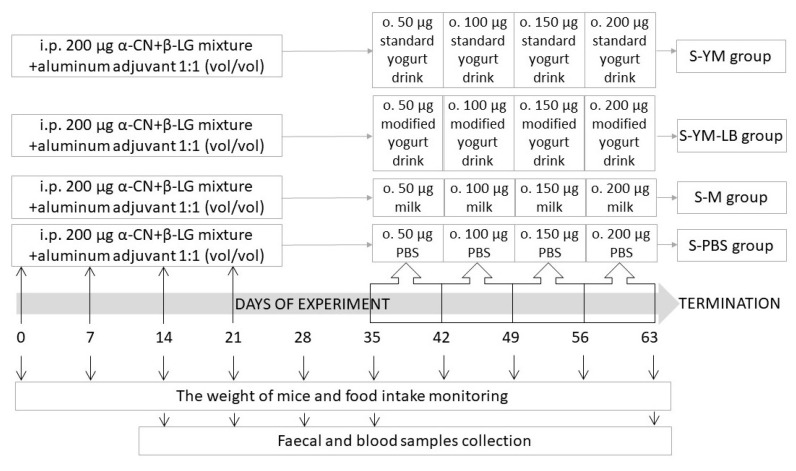
Scheme of the in vivo experiment—mice model. Mice groups: S-YM, mice sensitized via four intraperitoneal injection (i.p.) of a mixture of α-casein and β-lactoglobulin (α-CN + β-LG; 200 μg/100 μL) with aluminum adjuvant (1:1 (*vol*/*vol*), which were given intragastric (o.) increasing doses of yogurt drink YM fermented by *S. thermophilus* 2K and *L. bulgaricus* BK; S-YM-LB, sensitized mice, which were given yogurt drink YM-LB fermented by *S. thermophilus* 2K, *L. bulgaricus* BK, *B. lactis* Bi30 and *L. plantarum* W42; S-M, sensitized mice given milk; S-PBS, sensitized mice given phosphate-buffered saline (PBS).

**Figure 2 nutrients-12-03390-f002:**
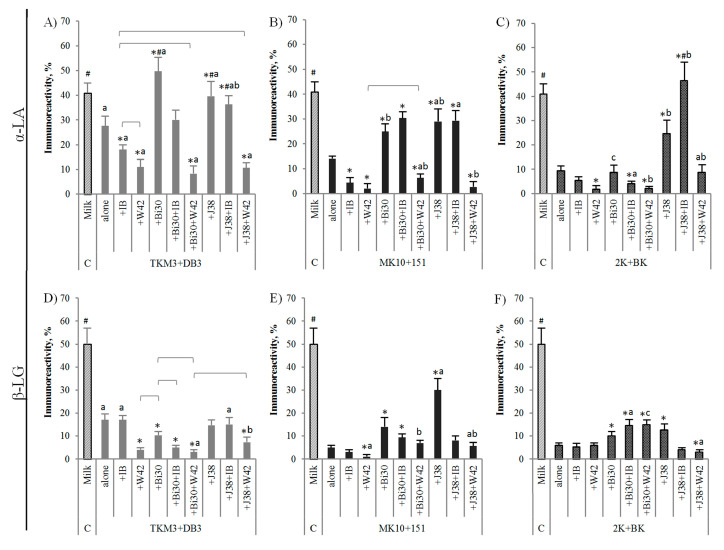
Immunoreactivity of whey proteins present in milk and in yogurt drinks fermented with different bacterial sets—ELISA results with anti-α-lactalbumin (anti-α-LA; graphs: **A**–**C**) and anti-β-lactoglobulin (anti-β-LG) antibodies (graphs: **D**–**F**). Strain names: TKM3, *Streptococcus salivarius* subsp. *thermophilus* TKM3; DB3, *Lactobacillus delbrueckii* subsp. *bulgaricus* DB3; IB, *Lactobacillus plantarum* IB; W42, *L. plantarum* W42; Bi30, *Bifidobacterium animalis* subsp. *lactis* Bi30; J38, *B. lactis* J38; MK-10, *S. thermophilus* MK-10; 151, *L. bulgaricus* 151; 2K, *S. thermophilus* 2K, BK, *L. bulgaricus* BK. For the detail explanation of strain compositions, see Table 1. The results are expressed as mean ± SD. Statistical analysis was performed by *t* test. * The means in the single graph are different from milk fermented with basic starter set (column “alone”) at *p* ≤ 0.05. ^#^ The means in the single graph without this superscript differ from unfermented milk (column “milk”) at *p* ≤ 0.05. The means marked with a buckle are different at *p* ≤ 0.05 (the comparison was performed only for the results which were significantly lower from the basic starter (column “alone”). The means for the same combination of added bacteria but different basic starter cultures used for milk fermentation (compared results are shown in three graphs in the horizontal row, i.e., **A**–**C** or **D**–**F**) marked with a letter or different letters differ at *p* ≤ 0.05.

**Figure 3 nutrients-12-03390-f003:**
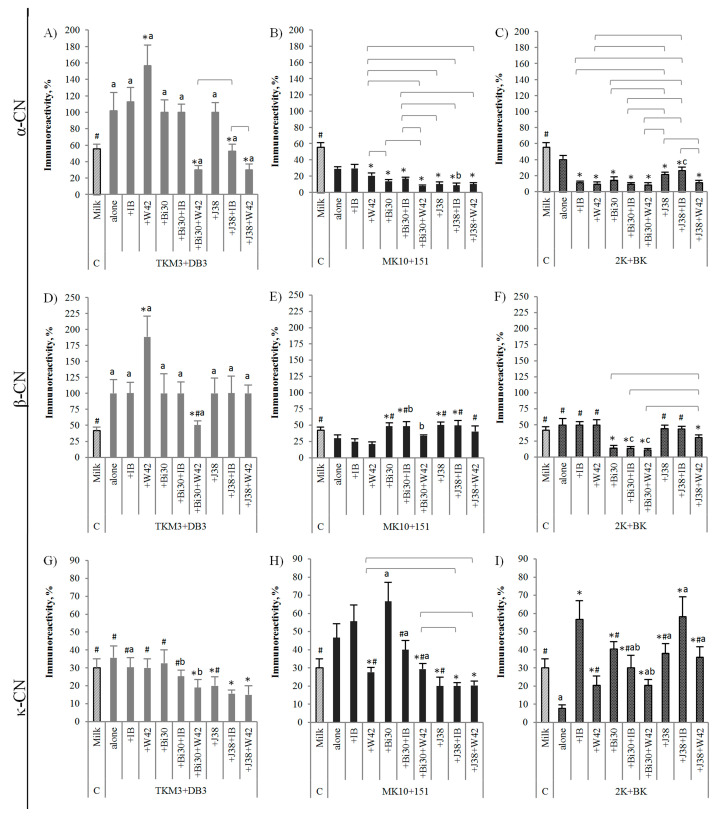
Immunoreactivity of casein proteins present in milk and in yogurt drinks fermented with different bacterial sets—ELISA results with anti-α-casein (anti-α-CN; graphs: **A**–**C**), anti-β-casein (anti-β-CN; graphs: **D**–**F**), anti-κ-casein (anti-κ-CN) antibodies (graphs: **G**–**I**). Strain names: TKM3, *Streptococcus salivarius* subsp. *thermophilus* TKM3; DB3, *Lactobacillus delbrueckii* subsp. *bulgaricus* DB3; IB, *Lactobacillus plantarum* IB; W42, *L. plantarum* W42; Bi30, *Bifidobacterium animalis* subsp. *lactis* Bi30; J38, *B. lactis* J38; MK-10, *S. thermophilus* MK-10; 151, *L. bulgaricus* 151; 2K, *S. thermophilus* 2K, BK, *L. bulgaricus* BK. For the detail explanation of strain compositions, see Table 1. The results are expressed as mean ± SD. Statistical analysis was performed by *t* test. * The means in the single graph are different from milk fermented with basic starter set (column “alone”) at *p* ≤ 0.05. ^#^ The means in the single graph without this superscript differ from unfermented milk (column “milk”) at *p* ≤ 0.05. The means marked with a buckle are different at *p* ≤ 0.05 (the comparison was performed only for the results which were significantly lower from the basic starter (column “alone”). The means for the same combination of added bacteria but different basic starter cultures used for milk fermentation (compared results are shown in three graphs in the horizontal row, i.e., **A**–**C**, **D**–**F**, or **G**–**I**) marked with a letter or different letters differ at *p* ≤ 0.05.

**Figure 4 nutrients-12-03390-f004:**
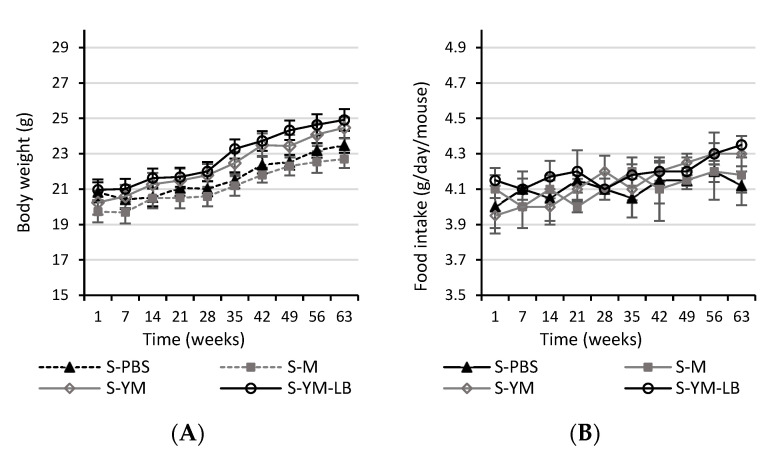
The body weight (**A**) and food intake (**B**) of the mice throughout the study. Mice groups: S-PBS—sensitized with a mixture of α-casein and β-lactoglobulin (α-CN + β-LG −200 μg of protein/100 μL of mixture) with aluminum adjuvant (1:1 *v*/*v*) and treated with phosphate-buffered saline (PBS), S-M—sensitized and treated with milk, S-YM—sensitized and treated with YM yogurt drink fermented with *S. thermophilus* 2K + *L. bulgaricus* BK starter set, S-YM-LB—sensitized and treated with YM-LB yogurt drink fermented with *B. lactis* Bi30 + *L. plantarum* W42 + *S. thermophilus* 2K + *L. bulgaricus* BK bacterial set. Food intake was measured per cage and calculated per mouse (n = 3 per group). Differences were analyzed with a one-way ANOVA follow by Tukey post-hoc test (**A**) or a Kruskal–Wallis test (**B**). Values are expressed as mean ± SEM.

**Figure 5 nutrients-12-03390-f005:**
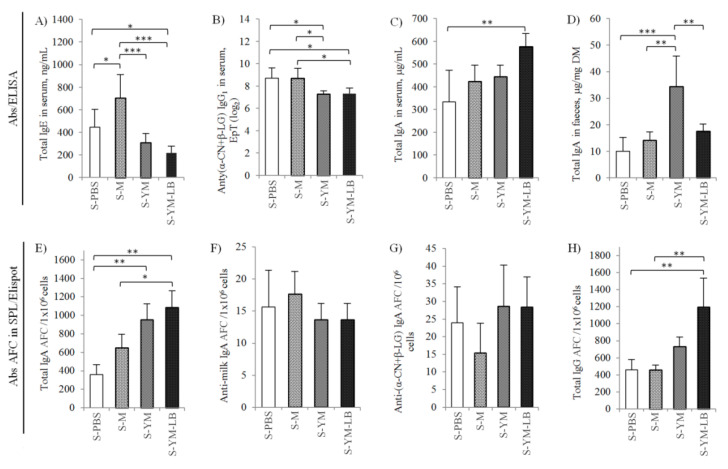
Humoral immune response of sensitized mice after four-week of experimental feeding. ELISA results: (**A**) total IgE in serum; (**B**) IgG_1_ specific to α-casein + β-lactoglobulin (α-CN + β-LG) in serum, terminal specific antibodies endpoint titer (EpT); (**C**) total IgA in serum; (**D**) total IgA in feces. ELISpot results: (**E**) total IgA antibody-forming cells (AFC) in splenocytes; (**F**,**G**) total IgA AFC specific to milk and α-CN + β-LG, respectively (in splenocytes); (**H**) total IgG AFC in splenocytes. Mice groups: S-PBS—sensitized with milk allergens (α-CN + β-LG −200 μg of protein/100 μL of mixture) with aluminum adjuvant (1:1 *v*/*v*) and treated with phosphate-buffered saline (PBS), S-M—sensitized and treated with milk, S-YM—sensitized and treated with the YM yogurt drink fermented with *S. thermophilus* 2K + *L. bulgaricus* BK starter set, S-YM-LB—sensitized and treated with YM-LB yogurt drink fermented *B. lactis* Bi30 + *L. plantarum* W42 + *S. thermophilus* 2K + *L. bulgaricus* BK bacterial set. Data are expressed as the mean ± SD. Statistical analysis was performed by one-way ANOVA with the post hoc Tukey test. The means marked with a buckle are different: * *p* ≤ 0.05, ** *p* ≤ 0.01, *** *p* ≤ 0.001.

**Figure 6 nutrients-12-03390-f006:**
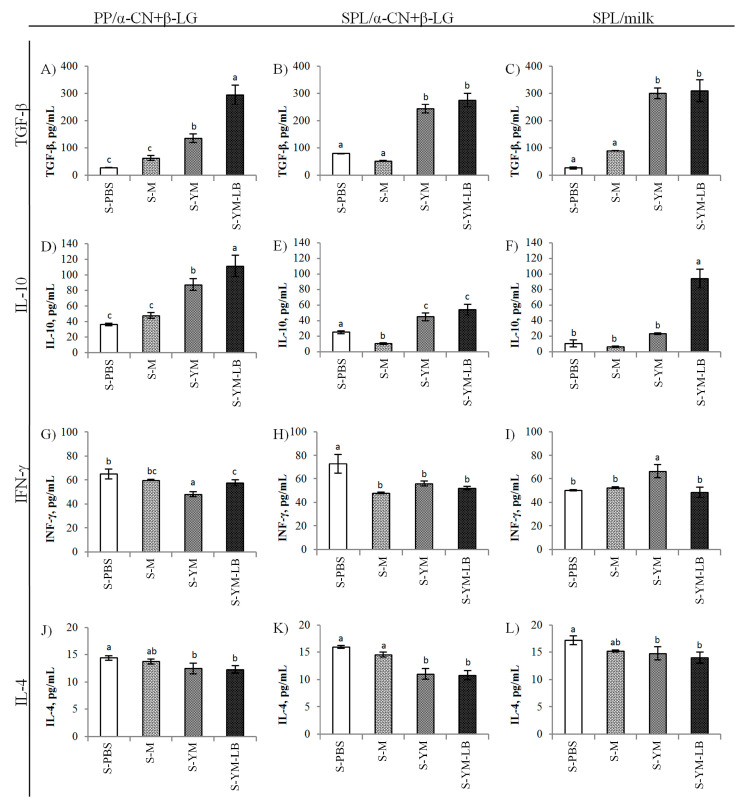
Cytokines secreted by lymphocytes isolated from the tested mice and co-cultured in vitro with α-casein + β-lactoglobulin (α-CN + β-LG) or milk. (**A**,**D**,**G**,**J**) lymphocytes isolated from Peyer’s patches (PPs) co-cultured with α-CN + β-LG; (**B**,**E**,**H**,**K**) lymphocytes isolated from spleens (SPL) co-cultured with α-CN + β-LG; (**C**,**F**,**I**,**L**) lymphocytes isolated from SPL co-cultured with milk. Mice groups: S-PBS—sensitized with milk allergens (α-CN + β-LG—200 μg of protein/100 μL of mixture) with aluminum adjuvant (1:1 *v*/*v*) and treated with phosphate-buffered saline (PBS), S-M—sensitized and treated with milk, S-YM—sensitized and treated with YM yogurt drink fermented with *S. thermophilus* 2K + *L. bulgaricus* BK starter set, S-YM-LB—sensitized and treated with YM-LB yogurt drink fermented *B. lactis* Bi30 + *L. plantarum* W42 + *S. thermophilus* 2K + *L. bulgaricus* BK bacterial set. Data are expressed as the mean ± SD. Statistical analysis was performed by one-way ANOVA with post hoc Tukey test. The means marked with different letters differ at *p* ≤ 0.05.

**Figure 7 nutrients-12-03390-f007:**
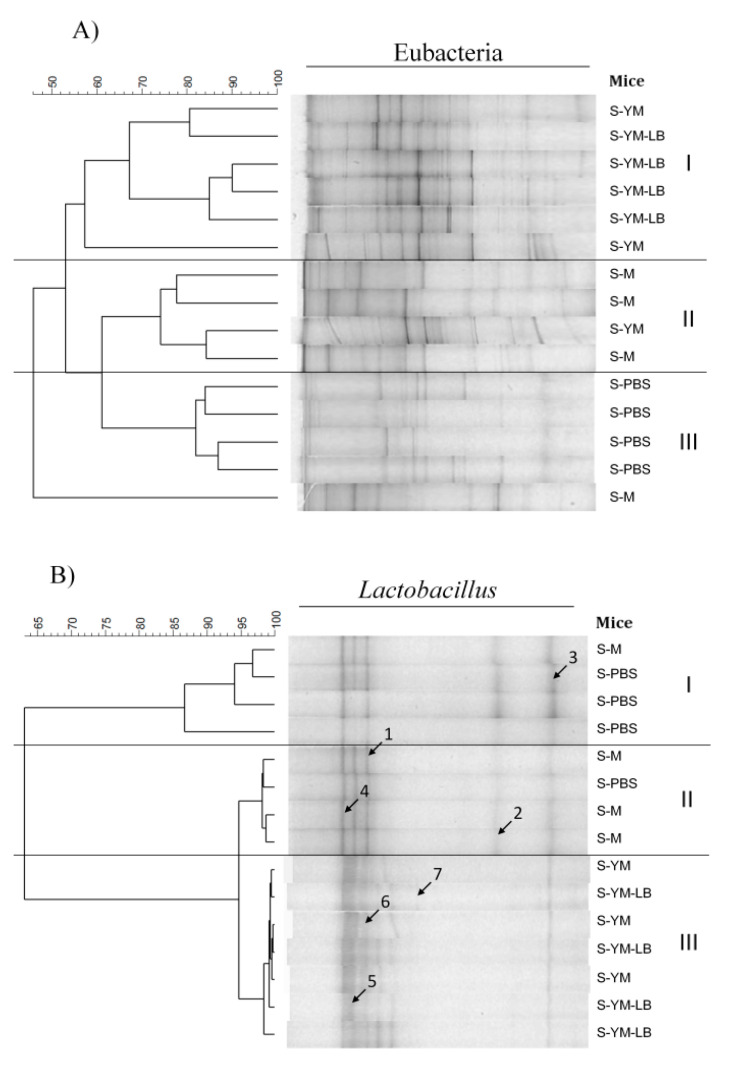
Profile of gut microbiota of the tested mice. Denaturing gradient gel electrophoresis (DGGE) banding patterns obtained with universal (**A**) and *Lactobacillus*-specific (**B**) primers. The comparison based on profile similarities calculated using the Pearson’s correlation coefficient, dendrogram constructed using unweighted pair group method with arithmetic mean (UPGMA). Mice groups: S-PBS—sensitized with a mixture of α-casein and β-lactoglobulin (α-CN + β-LG −200 μg of protein/100 μL of mixture) with aluminum adjuvant (1:1 *vol*/*vol*) and treated with phosphate-buffered saline (PBS), S-M—sensitized and treated with milk, S-YM—sensitized and treated with YM yogurt drink fermented with *S. thermophilus* 2K + *L. bulgaricus* BK starter set, S-YM-LB—sensitized and treated with YM-LB yogurt drink fermented *B. lactis* Bi30 + *L. plantarum* W42 + *S. thermophilus* 2K + *L. bulgaricus* BK bacterial set. Results of identification of the sequenced bands of lactobacilli: (band 1, 3)—*Lactobacillus reuteri* (98–100% sequence similarity), (band 2, 5, 6)—*Lactobacillus gasseri* (98%), (band 4)—*lactobacillus murinus* (98%), (band 7)—*Lactobacillus bulgaricus* (98).

**Table 1 nutrients-12-03390-t001:** Strain composition in the yogurt sets applied for milk fermentation.

Basic Starter	Added Bacteria	Strain Composition	Set Name
TKM3 + DB3	none (alone)	*S. thermophilus* TKM3, *L. bulgaricus* DB3	TKM3 + DB3
	+IB	*S. thermophilus* TKM3, *L. bulgaricus* DB3, *L. plantarum* IB	TKM3 + DB3 + IB
	+W42	*S. thermophilus* TKM3, *L. bulgaricus* DB3, *L. plantarum* W42	TKM3 + DB3 + W42
	+Bi30	*S. thermophilus* TKM3, *L. bulgaricus* DB3, *B. lactis* Bi30	TKM3 + DB3 + Bi30
	+J38	*S. thermophilus* TKM3, *L. bulgaricus* DB3, *B. lactis* J38	TKM3 + DB3 + J38
	+Bi30 + IB	*S. thermophilus* TKM3, *L. bulgaricus* DB3, *B. lactis* Bi30, *L. plantarum* IB	TKM3 + DB3 + Bi30 + IB
	+Bi30 + W42	*S. thermophilus* TKM3, *L. bulgaricus* DB3, *B. lactis* Bi30, *L. plantarum* W42	TKM3 + DB3 + Bi30 + W42
	+J38 + IB	*S. thermophilus* TKM3, *L. bulgaricus* DB3, *B. lactis* J38, *L. plantarum* IB	TKM3 + DB3 + J38 + IB
	+J38 + W42	*S. thermophilus* TKM3, *L. bulgaricus* DB3, *B. lactis* J38, *L. plantarum* W42	TKM3 + DB3 + J38 + W42
MK10 + 151	none (alone)	*S. thermophilus* MK-10, *L. bulgaricus* 151	MK10 + 151
	+IB	*S. thermophilus* MK-10, *L. bulgaricus* 151, *L. plantarum* IB	MK10 + 151 + IB
	+W42	*S. thermophilus* MK-10, *L. bulgaricus* 151, *L. plantarum* W42	MK10 + 151 + W42
	+Bi30	*S. thermophilus* MK-10, *L. bulgaricus* 151, *B. lactis* Bi30	MK10 + 151 + Bi30
	+J38	*S. thermophilus* MK-10, *L. bulgaricus* 151, *B. lactis* J38	MK10 + 151 + J38
	+Bi30 + IB	*S. thermophilus* MK-10, *L. bulgaricus* 151, *B. lactis* Bi30, *L. plantarum* IB	MK10 + 151 + Bi30 + IB
	+Bi30 + W42	*S. thermophilus* MK-10, *L. bulgaricus* 151, *B. lactis* Bi30, *L. plantarum* W42	MK10 + 151 + Bi30 + W42
	+J38 + IB	*S. thermophilus* MK-10, *L. bulgaricus* 151, *B. lactis* J38, *L. plantarum* IB	MK10 + 151 + J38 + IB
	+J38 + W42	*S. thermophilus* MK-10, *L. bulgaricus* 151, *B. lactis* J38, *L. plantarum* W42	MK10 + 151 + J38 + W42
2K + BK	none (alone)	*S. thermophilus* 2K, *L. bulgaricus* BK	2K + BK ^1^
	+IB	*S. thermophilus* 2K, *L. bulgaricus* BK, *L. plantarum* IB	2K + BK + IB
	+W42	*S. thermophilus* 2K, *L. bulgaricus* BK, *L. plantarum* W42	2K + BK + W42
	+Bi30	*S. thermophilus* 2K, *L. bulgaricus* BK, *B. lactis* Bi30	2K + BK + Bi30
	+J38	*S. thermophilus* 2K, *L. bulgaricus* BK, *B. lactis* J38	2K + BK + J38
	+Bi30 + IB	*S. thermophilus* 2K, *L. bulgaricus* BK, *B. lactis* Bi30, *L. plantarum* IB	2K + BK + Bi30 + IB
	+Bi30 + W42	*S. thermophilus* 2K, *L. bulgaricus* BK, *B. lactis* Bi30, *L. plantarum* W42	2K + BK + Bi30 + W42 ^2^
	+J38 + IB	*S. thermophilus* 2K, *L. bulgaricus* BK, *B. lactis* J38, *L. plantarum* IB	2K + BK + J38 + IB
	+J38 + W42	*S. thermophilus* 2K, *L. bulgaricus* BK, *B. lactis* J38, *L. plantarum* W42	2K + BK + J38 + W42

^1^ This set was applied to produce yogurt drink YM for the in vivo experiment. ^2^ This set was applied to produce yogurt drink YM-LB for the in vivo experiment. Strain names: TKM3, *Streptococcus salivarius* subsp. *thermophilus* TKM3; DB3, *Lactobacillus delbrueckii* subsp. *bulgaricus* DB3; IB, *Lactobacillus plantarum* IB; W42, *L. plantarum* W42; Bi30, *Bifidobacterium animalis* subsp. *lactis* Bi30; J38, *B. lactis* J38; MK-10, *S. thermophilus* MK-10; 151, *L. bulgaricus* 151; 2K, *S. thermophilus* 2K, BK, *L. bulgaricus* BK.

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
