# Peer review of "Effect of Low-Immunogenic Yogurt Drinks and Probiotic Bacteria on Immunoreactivity of Cow’s Milk Proteins and Tolerance Induction—In Vitro and In Vivo Studies"

_nutrients, 2020, doi:10.3390/nu12113390_

Round 1

Reviewer 1 Report

The authors present their manuscript looking at an experimental model of food allergy.  They look at the affect of different bacterial strains and yogurt preparation on the effect of an experimental mouse model of food allergy. 

Minor comments

Authors should be aware of abbreviations.  first time things are used they need to be written out before abbreviation even in the abstract.

Manuscript would be improved with a figure that shows the experimental model and the divided groups

Manuscript need to be more concise, the authors present a lot of work but the flow of the article is dense and hard to follow

Results and Discussion should be separated. With the discussion built into the result sections get very long with no conclusion

Does the microbiota study need to be in this paper?  How does it add to the study?

Reviewer 2 Report

The paper investigates the combination of lactic acid bacteria and probiotics for the yogurt drinks as low-risk and immunomodulatory foods for the patients having cow’s milk allergy. The paper is helpful for the progress of the yogurt drinks for the treatment of CMA patients. Still, I have small questions to be addressed.

P1 L16 and P2 L82 

Would you please explain the abbreviation "CN"?

P6 L193 

the submandibular "vein"

P17 L607 and Sumary

The authors concluded that the immune response inclined toward Th1-type response. However, both IL-4 and IFN-g declined by the yogurt intakes in Figure5. The authors mentioned that IFN-g does not necessarily represent the Th1-type immunity... From which results did the authors conclude the shift of the Th2 to the Th1-immunity? Could you specify it?

P17 L602

The authors said that the tested yogurt drinks mitigated inflammatory reactions.

However, it is unclear because the inflammatory symptoms in the models were not mentioned in the manuscript. Does it mean the alleviation of the skin reactions or body temperature changes in the models? Or does it mean the changes in the profiles of antibody and cytokine responses? Could you explain it?
